Comparison between radial artery tonometry pulse analyzer and pulsed-Doppler echocardiography derived hemodynamic parameters in cardiac surgery patients: a pilot study

Zayat Rashad rzayat@ukaachen.de r.zayat@gmail.com 1
Goetzenich Andreas 1
Lee Ju-Yeon 2
Kang HeeJung 3
Jansen-Park So-Hyun 2
Schmitz-Rode Thomas 2
Musetti Giulia 1
Schnoering Heike 1
Autschbach Rüdiger 1
Hatam Nima 1
Aljalloud Ali 1
1 Department of Thoracic and Cardiovascular Surgery, RWTH University Hospital Aachen , Aachen , Germany
2 Institute of Applied Medical Engineering, Helmholtz Institute, RWTH Aachen University , Aachen , Germany
3 DAEYOMEDI Co. Ltd. , Ansan-Si, Gyeonggi-do , South Korea
Ma Shuangtao
Electronic publication date: 2017 Dec 6
Publication date: 2017
Volume: 5
Electronic Location ID: e4132
Received 2017 Sep 13; Accepted 2017 Nov 14
Copyright: ©2017 Zayat et al.
Copyright year: 2017
Copyright holder: Zayat et al.
License: This is an open access article distributed under the terms of the Creative Commons Attribution License, which permits unrestricted use, distribution, reproduction and adaptation in any medium and for any purpose provided that it is properly attributed. For attribution, the original author(s), title, publication source (PeerJ) and either DOI or URL of the article must be cited.
License URL: https://creativecommons.org/licenses/by/4.0/

Keywords: Cardiac output, Stroke volume, Pulsed wave doppler, Radial artery, Tonometry, Transthoracic echocardiography

Funding: Department of Thoracic and Cardiovascular Surgery This work was supported by departmental (Department of Thoracic and Cardiovascular Surgery) resources. The funders had no role in study design, data collection and analysis, decision to publish, or preparation of the manuscript.

==============================
Background

Bedside non-invasive techniques, such as radial artery tonometry, to estimate hemodynamic parameters have gained increased relevance as an attractive alternative and efficient method to measure hemodynamics in outpatient departments. For our pilot study, we sought to compare cardiac output (CO), and stroke volume (SV) estimated from a radial artery tonometry blood pressure pulse analyzer (BPPA) (DMP-Life, DAEYOMEDI Co., Gyeonggi-do, South Korea) to pulsed-wave Doppler (PWD) echocardiography derived parameters.

Methods

From January 2015 to December 2016, all patients scheduled for coronary artery bypass (CABG) surgery at our department were screened. Exclusion criteria were, inter alia, moderate to severe aortic- or Mitral valve disease and peripheral arterial disease (PAD) > stage II. One hundred and seven patients were included (mean age 66.1 ± 9.9, 15 females, mean BMI 27.2 ± 4.1 kg/m2). All patients had pre-operative transthoracic echocardiography (TTE). We measured the hemodynamic parameters with the BPPA from the radial artery, randomly before or after TTE. For the comparison between the measurement methods we used the Bland-Altman test and Pearson correlation.

Results

Mean TTE-CO was 5.1 ± 0.96 L/min, and the mean BPPA-CO was 5.2 ± 0.85 L/min. The Bland-Altman analysis for CO revealed a bias of −0.13 L/min and SD of 0.90 L/min with upper and lower limits of agreement of −1.91 and +1.64 L/min. The correlation of CO measurements between DMP-life and TTE was poor (r = 0.501, p < 0.0001). The mean TTE-SV was 71.3 ± 16.2 mL and the mean BPPA-SV was 73.8 ± 19.2 mL. SV measurements correlated very well between the two methods (r = 0.900, p < 0.0001). The Bland-Altman analysis for SV revealed a bias of −2.54 mL and SD of ±8.42 mL and upper and lower limits of agreement of −19.05 and +13.96 mL, respectively.

Conclusion

Our study shows for the first time that the DMP-life tonometry device measures SV and CO with reasonable accuracy and precision of agreement compared with TTE in preoperative cardiothoracic surgery patients. Tonometry BPPA are relatively quick and simple measuring devices, which facilitate the collection of cardiac and hemodynamic information. Further studies with a larger number of patients and with repeated measurements are in progress to test the reliability and repeatability of DMP-Life system.

Introduction

Hemodynamic monitoring is crucial, not only during anesthesia and in intensive care units (ICU) but also in the normal ward and in the follow-up during outpatient visits, especially in cardiac surgery patients. The main goal of hemodynamic monitoring is to ensure a sufficient end organ perfusion and oxygen delivery by optimizing stroke volume (SV) and cardiac output (CO). The gold standard in monitoring CO and SV is still the pulmonary artery catheter (PAC), its invasive nature and life-threatening complications largely restrict its use in the operating room and in the ICU (Marik, 2013; Wheeler et al., 2006). During the last decade, several non- or minimally invasive techniques have been presented, such as pulse wave transit time, non-invasive pulse contour analysis, arterial tonometry, oscillometry, esophageal Doppler devices, the partial carbon dioxide rebreathing technique, and transthoracic electrical bio-impedance measurements. However, PAC is still the most accepted method as there is still no clearly established gold standard for CO measurement in human studies (Peyton & Chong, 2010; Sato et al., 1993; Thiele, Bartels & Gan, 2015). Concurrently, both transesophageal (TEE) and transthoracic (TTE) echocardiography, have become a frequently utilized monitor in the cardiac operating rooms and ICU. TEE/TTE derived SV and CO have been validated against PAC-based values (Harris, Luther & Perrino, 1999; Perrino Jr, Harris & Luther, 1998). TTE/TEE have many advantages, but there are some limitations, such as the need for an accurate ultrasound window, which is not always possible post sternotomy, semi-continuous real-time monitoring, and operator dependence. Lately, radial artery tonometry with the use of a piezoresistive array sensor has been presented as a non-invasive method to monitor the blood pressure and analyze the radial artery waveform (Jun et al., 2016; Shin et al., 2010). In this pilot study, we aimed to compare the CO and SV estimated from a new radial tonometry blood pressure pulse analyzer (BPPA) using the piezoresistive array sensor array technique (DMP-Life, DAEYOMEDI Co., Ltd., Gyeonggi-do, South Korea) with the TTE derived CO and SV in cardiac surgery patients prior to surgery.

Materials and Methods

Patients and data collection

Our study was conducted in the cardio-surgical department of a German university hospital (RWTH University Hospital, Aachen, Germany). The ethics committee (Ethikkommission der RWTH Aachen) approved the study (EK 151/90). Written informed consent was obtained from all patients prior to study enrolment.

Between January 2015 and December 2016, all patients who were planned for nonemergency coronary artery bypass graft (CABG) surgery, were screened. Exclusion criteria were: (1) peripheral artery disease (PAD) more than Fontaine grade II, (2) moderate to severe aortic stenosis or regurgitation, (3) moderate to severe mitral stenosis or regurgitation, (4) atrial fibrillation, (5) left ventricular ejection fraction <40%, (6) no accurate ultrasound window for TTE. Five hundred and seventy-two patients were screened and 107 patients were included in the study (Fig. 1). Patients’ data were prospectively entered to our institution’s electronic database.

Figure 1 Study flow chart.

LVEF, Left ventricular ejection fraction; PAD, Peripheral artery disease; TTE, transthoracic echocardiography.

Echocardiographic and tonometry measurements

After enrolment in the study, all patients had TTE examinations 2 to 3 days prior to the scheduled CABG. TTE studies were performed using a commercially available machine (Vivid E9® Vingmed-General Electrics, Horten, Norway), and data analysis was performed offline using the Echopac system. TTE were performed according to the American society of echocardiography and European association for echocardiography (ASE/EAE) guidelines (Lang et al., 2015). All TTE studies and measurements with the DMP-Life were performed in the same room, under standardized conditions and by the same operators for the entire cohort. TTE examinations were performed in a left lateral position. Measurements with DMP-life were obtained in a lying supine position from the left or right radial artery, randomly. The average of DMP-life measurements from at least five heart beats were calculated. The two measurements (TTE and DMP-life) were performed in random order.

TTE derived SV was then calculated using the pulsed Doppler (PW) through the left ventricular outflow tract (LVOT) as a product of the velocity time integral through the LVOT and the cross-sectional area of the LVOT as described by Lewis et al. (1984). The stroke volume (SV) was calculated by measuring: (1) the diameter (d) of the LVOT from the parasternal window was used to calculate the cross-sectional area (CSA) (CSALV OT = (d∕2)2 × π); (2) the velocity time integral (VTI) measured at the same site (LVOTV TI) at the apical three-chamber view with pulsed-wave Doppler. SV averaged over five consecutive beats was used to compute the CO (SV × heart rate).

DMP-Life system description

The pulse was measured by radial artery tonometry device DMP-Life (DAEYOMEDI Co., South Korea). With the array pressure sensor equipment, tonometry devices have improved to confirm blood vessels’ position accurately and also improved reproducibility and reliability. Through orthogonal applanation to radial artery, tonometry devices can get signals, which include blood pressure, skin condition, blood vessel stiffness and hemodynamic condition (Jun et al., 2016; Singh et al., 2017; Wagner et al., 2016).

The sensor module of DMP-Life is equipped with multi-channel array with five piezoresistive semiconductor transducers (Fig. 2A) and precise moving actuator to apply pressure automatically by a given algorithm. During pressure application on radial artery, the actuator moves very slowly to avoid any fluctuation on the signal baseline, to have a maximum control resolution of 12 µm, and to detect accurate skin property and blood pressure (Fig. 2C).

Figure 2 DMP-Life System.

(A) Dorsal view of the DMP-life system, with the array sensor in the middle of the housing. The arrows indicate the position of the five piezoresistive sensors. (B) Cell diagram of the semiconductor pressure sensor: DMP-Life has multi cells in one sensor tip, one sensor cell has four resistance materials (R1, R2, R3, R4). 1–6 are the connecting points of the resistance material. One sensor cell gives an output of one pressure result. (C) Schematic illustration of the sensor with the actuator while applying the pressure slowly on the radial artery to detect the optimal required pressure. (D) Demonstration of how the DMP-Life is positioned on the left wrist. For further description please refer to the main document under measurements with the DMP-life system.

The range of the applied pressure in DMP-Life system is set at 0∼400 g f/cm2, but for each patient, individual maximum applied pressure can vary according to patient’s pulse pressure response. Waveforms at each applying pressure can be used to analyze the elasticity and stiffness of the vascular system. The concept of pulse pressure (systolic blood pressure minus the diastolic pressure) is applied to the tonometric sensor.

In DMP-life, the difference between the peak voltage and the lowest voltage measured in one cycle is defined as a pulse force. Figure 3 shows some cases of measured pulse forces for each applying pressure. For example, in Fig. 3A the maximum pulse height in this patient is reached when a force of 150 g f/cm2 is applied, and in Fig. 3D the patient’s maximum pulse height is first reached when a pressure of 260 g f/cm2 was applied. The response of the pulse to the applied pressure result in a typical shape for each patient. Sharp end or arrow shape or triangle shape may appear in healthy elastic vessel (Figs. 3A , 3B and 3D), but smooth and flat curved end similar to half-arc shape may result due to stiff or tensed vessel (Fig. 3C). The correlation of different shapes due to individual vessels response with the severity of cardiovascular disease is still under investigation.

Figure 3 Exemplary demonstration of the range of the applied pressure and the corresponding vessels response.

Each yellow dot is the value of pulse height vs. applied pressure. The yellow dots make a typical shape of vessel response for each patient. Sharp end or arrow shape (like a triangle) may occur in healthy elastic vessel (A, B and D), but smooth and flat curved end similar to a half-arc shape may result due to stiff and tensed vessel (C). The applied pressure differs in each patient according to the vessel’s response to the applied pressure. (A–D) demonstrate the detection of the maximum pulse height at different applied pressure for each person (in A maximum pulse height is reached when applying 150 g f/cm2, in B when applying 150 g f/cm2, in C maximum reach when applying 125 g f/cm2, in D when applying 260 g f/cm2). (E) shows fully measured pulse data with stable baseline. The red rectangle shows the average range of applied pressure and the corresponding pulse pressure changes in healthy people. The red arrow indicates the increase of applied pressure (from right to left).

The multi-channel array with five piezoresistive scan the radial pulse location to identify the site of maximum pulsation. The pulse pressure amplitude varies least at distances furthest from the vessel to greatest at the optimal position over the pulse. Thick skin, deep arteries and high mean artery blood pressure can be a cause of stronger pressure and long measurement time, because the actuator moves in constant velocity during detection of the pulse signal. The measured pulse signal is analyzed by a patented algorithm, which is a modified algorithm based on the systolic area with Kouchoukos correction algorithm (Kouchoukos, Sheppard & McDonald, 1970) and the algorithm hold a patent from the Korean Intellectual Property Office (Patent number: 100785901000, 1006948960000).

The measured data with the DMP-Life system are: systolic blood pressure (SBP), diastolic blood pressure (DBP), pulse pressure (PP), pulse rate (PR), radial-augmented index (R-AI), stroke Volume (SV), stroke volume index (SVI), cardiac output (CO), cardiac index (CI), systemic circulation resistance index (SCRI) and pulse conditions. For this pilot study, we used only SV and CO.

DMP-Life uses a modified systolic area with Kouchoukos correction algorithm (Kouchoukos, Sheppard & McDonald, 1970) to calculate SV with measured waveform parameter. SV  = a + b × (T4 + c) + d × PR + e × PP + f × BSA + g × Age; and CO = SV × PR, where, T4: LV (sec); PR: Pulse rate (beat/min); PP: Pulse Pressure (mmHg); BSA: Body surface area (m2); a, b, c, d, e, f and g: coefficient constants. DMP-life has tonometric blood pressure measure function and does not need calibration with other blood pressure monitor.

Measurements with the DMP-life system

Before performing the measurement with DMP-Life, patients should take at least 5 min of rest time. Patients were in lying supine position. After entering the patient’s biometric data (height, weight, age, and gender), the physician then first felt the radial artery pulse and chose the best position. A bracelet with the pressure sensor cartridge (DMP-life) was placed on the patient’s wrist over the radial artery (Fig. 2D). The sensor of DMP-life includes a multi-channel array with five piezoresistive semiconductor transducers (Fig. 2A). When the physician turns the device on, the device software automatically judges and gives notice to the operator whether the sensor module is positioned well or not. When the sensor position is confirmed, the actuator moves in constant velocity and applies pressure on the radial artery automatically by a given algorithm to partially flatten the radial artery (Fig. 2C). The radial artery pressure is then transmitted from the vessel to the sensor and is recorded digitally. When the optimal signal is available, the DMP-life monitor provides a continuous arterial waveform (Fig. 4A). The quality of the captured pulse waves can be visually assessed by the physician on the monitor and also the device performs default quality check. (1) The system automatically confirms vessel position when the array sensor starts capturing the pulse waves. If the sensor position is out of vessel’s area then the system gives a note and ask the operator to repeat the measurements. (2) The system automatically detect applied pressure and corresponding changes in pulse waves. If inappropriate pulse wave signals are captured with abnormal vessels respond, the system quits measuring and asks for repeating the measurements.

Figure 4 Exemplary demonstration of the results on the DMP-life monitor.

(A) When the optimal signal is available, the DMP-life monitor provides a continuous arterial waveform. (B) The DMP-life system’s monitor displays inter alia CO and SV values derived from pulse contour analysis. ESV, estimated stroke Volume (ESV); ESVI, estimated stroke volume index; ECO, estimated cardiac output; ECRI, estimated systemic circulation resistance index; SBP, systolic blood pressure; DBP, diastolic blood pressure.

The DMP-life system’s monitor displays inter alia CO and SV values derived from pulse contour analysis using a proprietary auto-calibrating algorithm, as mentioned above (Fig. 4B).

Statistical analysis

Continuous variables are presented as mean ± standard deviation (SD) or as median (25th and 75th percentile), where appropriate. Categorical variables are described in absolute numbers and percentages. We computed the mean of the differences (=bias) between BPPA- and TTE-CO/SV, the SD, and the 95% limits of agreement (=bias ±1.96 × SD) to describe the agreement between BPPA (DMP-life) and TTE measurements, Bland–Altman plots for repeated measures were also calculated (Bland & Altman, 2007). The Pearson product moment correlation test was used to evaluate the correlation of SV and CO between the two methods (DMP-life and TTE). All statistical analyses were performed using IBM SPSS Statistics 23 (SPSS, Chicago, IL, USA).

Results

Patients characteristics

One hundred and seven patients were included in the study. The mean age was 66.1 ± 9.9 years and 14.1% were female. Clinical characteristics of the patients are demonstrated in Table 1. Drug use was: beta blockers (96%), angiotensin-converting enzyme inhibitors and/or angiotensin 2 receptor blockers (85%), diuretics (78%), spironolactone (10%), statins (68%) and aspirin (98%), respectively.

Table 1 Patients’ demographic information.

Variables	Patients (n = 107)	
Age years	67 (58, 73)	
Female n (%)	15 (14.1%)	
BMI kg/m2	26.7 (24.5, 29.7)	
Peripheral artery disease I–II°n (%)	22 (20.6)	
Diabetes mellitus n (%)	34 (31.8)	
Arterial hypertension n (%)	98 (91.6)	
Chronic obstructive lung disease n (%)	18 (16.8)	
Chronic kidney disease n (%)	20 (18.7)	
Left ventricular ejection fraction >50% n (%)	48 (44.8)	
Left ventricular ejection fraction 40–50% n (%)	59 (55.2)	
Mean Heartrate per minute	72 (64, 86)	
Mean systolic pressure mmHg	134 (128, 147)	
Mean diastolic pressure mmHg	83(78, 87)	
EuroSCORE II%	3.4 ± 1.9	
Notes.

BMI Body mass-index

EuroSCORE II European System for Cardiac Operative Risk Evaluation

SV and CO measurements

At the time of performing the measurements with TTE and DMP-life, the mean systolic blood pressure was 136.9 ± 11.6 mmHg, mean diastolic blood pressure was 82.2 ± 6.2 mmHg and the heart rate was 71.8 ± 15.1/min.

A total of 107 matched SV and CO data points were available for the final statistical analysis.

The mean TTE-SV was 71.3 ± 16.2 mL and the mean BPPA-SV was 73.8 ± 19.2 mL. The Bland-Altman analysis for SV revealed a bias of −2.54 mL and SD of ±8.42 mL and upper and lower limits of agreement of −19.05 and +13.96 mL, respectively (Fig. 5A). SV measurements correlated very well between the two methods, DMP-life and TTE, (r = 0.900, p < 0.0001) and the r squared (r2) for the goodness of fit was 0.811 (Fig. 5B). The mean TTE-CO was 5.1 ± 0.96 L/min. and the mean BPPA-CO was 5.2 ± 0.85 L/min. The Bland-Altman analysis for CO revealed a bias of −0.13 L/min and SD of 0.90 L/min with upper and lower limits of agreement of −1.91 and + 1.64 L/min (Fig. 5C). Pearson correlation demonstrated a poor correlation of the measured CO in the two methods (r = 0.501, p < 0.0001, r2 = 0.251) (Fig. 5D).

Figure 5 Bland-Altman and Pearson’s correlation plots.

(A) Bland-Altman plots of stroke volume (SV) measurements; (B) Pearson’s correlation of SV *; (C) Bland-Altman plots of cardiac output (CO); (D) Pearson’s correlation of CO measurements obtained from DMP-life and from transthoracic echocardiography in 107 patients. The dotted horizontal green line shows the mean of the differences (=bias) between the two methods, and the doted red horizontal lines show the upper and lower 95% limits of agreement (= bias ± 1.96 × SD). r: Correlation coefficient; r2: Squared r for the goodness of fit.

Discussion

Our study shows for the first time that the DMP-life technology measures SV and CO with reasonable accuracy and precision of agreement compared with TTE in preoperative cardiothoracic surgery patients. Our findings are promising as there is a clear trend toward non-invasive hemodynamic monitoring (Saugel & Reuter, 2014; Vincent et al., 2011). The bias for SV and CO between DMP-life and TTE were −2.54 mL and −0.13 L/min, respectively, which is acceptable in the clinical context but definitive recommendations for the definition of clinical acceptable agreement between two CO measuring systems are still under debate (Saugel & Reuter, 2014; Wagner et al., 2015). The Pearson correlation coefficient was excellent for the SV.

On the contrary, CO correlation was poor. As CO is calculated from SV and heart rate, this discordance can only be explained by a variation in patient’s heart rate. As TTE and DMP-life were not performed simultaneously, variation in the heart rate was present in all patients, explaining this result. PAC is still used as the reference method to monitor hemodynamics and to validate alternative monitoring systems (Rajaram et al., 2013), but its invasive nature and life-threatening complications largely restrict its use in the operating room and in the ICU (Marik, 2013; Wheeler et al., 2006). Echocardiography has been well accepted as a diagnostic tool for circulatory failure and as an alternative for PAC to evaluate hemodynamic parameters (De Backer, 2014; Porter et al., 2015). Measuring hemodynamics with TTE has some limitations: the need for a trained sonographer, poor acoustic window, inaccurate diameter calculations, and difficulty maintaining the angle of insonation (Expert Round Table on Echocardiography in ICU, 2014; Narasimhan, Koenig & Mayo, 2014). Although the PAC still the gold standard, echocardiography has been used to test newer hemodynamic monitoring systems (Gola et al., 1996; Romagnoli et al., 2013; Scolletta et al., 2016) and it has been recommended by international consensus as a reliable method for CO estimation (Cecconi et al., 2014). During the last decade, several non- or minimal invasive techniques have been presented, such as pulse wave transit time, non-invasive pulse contour analysis, arterial tonometry, oscillometry, oesophageal Doppler devices, the partial carbon dioxide rebreathing technique, and transthoracic electrical bio-impedance (Peyton & Chong, 2010; Sato et al., 1993; Thiele, Bartels & Gan, 2015). Our results and comparison are similar to De Castro et al. (2006), who used aortic Doppler echocardiography to compare the estimated SV from an axillary artery pulse-contour and found that there is a good correlation between pulse-contour analysis and aortic Doppler from transesophageal echocardiography (r = 0.842, p < 0.0001). Romagnoli et al. (2013) also analyzed TTE-CO in comparison with two different pulse contour devices. In a multicenter study by Scolletta et al. (2016) Doppler TTE was used as a comparison method to pulse contour device, and they could demonstrate a good correlation between TTE-CO and the CO estimated from a pulse contour device (r = 0.85; p < 0.0001). Our findings are in line with Wagner et al. (2015) who demonstrated that the applanation tonometry technology provides CO values with reasonable accuracy and precision of agreement compared with intermittent pulmonary artery thermodilution measurements. The mean of differences in the study by Wagner et al. (2015) was −0.2 L/min with 95% limits of agreement of −1.8 to + 1.4 L/min. The percentage error was 34%.

On the other hand Compton et al. (2008) compared radial artery tonometry to invasive estimated CO with PICCO® transpulmonary thermodilution or with PAC in 49 critically ill medical ICU patients and they concluded that radial artery applanation tonometry is not suitable to determine CO in critically ill hemodynamically unstable patients.

Arterial tonometry is a non-invasive method for blood pressure measurement and it provides complete pulse pressure waveform, which in turn has implications in several disease diagnoses (Laurent et al., 2006; Nelson et al., 2010).

A typical arterial tonometer consists of a plunger and pressure sensor placed at the centre of a superficial artery. The plunger is used for arterial flattening via the application of hold down pressure. A pressure sensor is used to measure contact stresses at the surface above the radial artery at optimal level of flattening (Singh et al., 2017). Different types of sensing elements in tonometer systems for effective pulse measurements and analyses, such as piezoelectric, piezoresistive, capacitive, and hall sensors, have been recently presented (Hu et al., 2012; Liu & Tyan, 2010). DMP-Life is equipped with multi-channel array with five piezoresistive semiconductor transducers and with a precise moving actuator to apply pressure automatically by specified algorithm. While applying the pressure on a radial artery, the actuator moves very slowly to avoid any fluctuation on signal baseline and to provide maximum control resolution with 12 µm to detect accurate skin properties and blood pressure (Fig. 2C). The piezoresistive sensor is the most precise method for measuring radial artery pulses because it can collect static and dynamic information on pulse waves with high sensitivity (Jun et al., 2016). The interaction between the shape of the plunger and the geometry of the wrist and the arrangement of the sensor array play a key role in precisely collecting and analyzing pulse wave signals (Jun et al., 2016; Singh et al., 2017). The concave shape of the DMP-life’s plunger and the arrangement of five piezoresistive sensors provide accurate interaction with the wrist geometry and allows estimating the optimal blood vessel direction as, when a sensor is at the center of the radial artery, it has a high-amplitude signal, and the signal becomes smaller as a sensor is far from the center. The DMP-Life filters automatically the signals from all five channels and chooses the optimum signal. Our stroke volume estimation is based on blood pressure, body surface area, and systolic time.

The DMP-Life pressure pulse analyzer is a relatively quick and simple measuring method, which facilitates the collection of cardiac and hemodynamic information. Since the tonometry technique of estimating the CO and SV is relatively accurate, it seems to be a good tool for patients with cardiac disease who need continuous self-monitoring.

Limitations of the Study

Our study is limited by the usual shortcomings of a small cohort single-center study and a probably heterogeneous patient group. In our proof of concept study, we did not use PAC as invasive method to compare CO and SV estimated from DMP-Life, which may limit the interpretation of the accuracy of DMP-life measurements, on the other hand, TTE measurements have been validated against PAC based values. A major issue related with the accuracy of CO/SV measurements is the reproducibility of the measurement, which was not assessed in our study and no repeated measurements in pre-, peri- and postoperative period were performed to determine whether DMP-Life detect changes in CO precisely. Another important issue was that measurements with DMP-life and TTE were not performed simultaneously, which did result in a poor correlation of CO, most likely due to different inter-individual heart rate at the time of performing the measurements.

One of the limitation of tonometry method is that the signal is very position sensitive, the transducer needs to be positioned directly over the center of the artery. This has been dealt with by using an array of five piezoresistive transducers placed across the artery.

Conclusion

Our study demonstrates for the first time that the DMP-life tonometry device measures SV and CO with reasonable accuracy and precision of agreement compared with TTE in preoperative cardiothoracic surgery patients. Tonometry BPPAs are relatively quick and simple measuring devices, which facilitates the collection of hemodynamic information. Further studies with a larger number of patients with repeated measurements are in progress to test the reliability and repeatability of DMP-Life system measurements.

Supplemental Information

Data S1 Raw data

Click here for additional data file.

Additional Information and Declarations

Competing Interests

Author Contributions

Human Ethics

Patent Disclosures

HeeJung Kang is an employee of DAEYOMEDI Co. Ltd.

Rashad Zayat conceived and designed the experiments, performed the experiments, analyzed the data, wrote the paper, prepared figures and/or tables.

Andreas Goetzenich conceived and designed the experiments, analyzed the data, contributed reagents/materials/analysis tools, prepared figures and/or tables.

Ju-Yeon Lee conceived and designed the experiments, performed the experiments, analyzed the data.

HeeJung Kang analyzed the data, contributed reagents/materials/analysis tools.

So-Hyun Jansen-Park performed the experiments, contributed reagents/materials/analysis tools, patients inclusion/exclusion.

Thomas Schmitz-Rode and Rüdiger Autschbach contributed reagents/materials/analysis tools, reviewed drafts of the paper.

Giulia Musetti performed the experiments.

Heike Schnoering contributed reagents/materials/analysis tools, reviewed drafts of the paper, patients inclusion/exclusion.

Nima Hatam conceived and designed the experiments, performed the experiments, analyzed the data, prepared figures and/or tables, reviewed drafts of the paper, critically revised the manuscript.

Ali Aljalloud conceived and designed the experiments, performed the experiments, analyzed the data, wrote the paper, reviewed drafts of the paper, critically revised the manuscript.

The following information was supplied relating to ethical approvals (i.e., approving body and any reference numbers):

The ethics committee (Ethikkommission der RWTH Aachen) approved the study (EK 151/90).

The following patent dependencies were disclosed by the authors:

The algorithm to calculate cardiac output from DMP-Life device is patented by the Korean Intellectual Property Office (Patent number: 100785901000, 1006948960000).

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
