# Peer review of "Comparison between radial artery tonometry pulse analyzer and pulsed-Doppler echocardiography derived hemodynamic parameters in cardiac surgery patients: a pilot study"

_PeerJ, doi:10.7717/peerj.4132_

## Round 0.1 · original submission · Major Revisions

We have 4 reviewers who carefully reviewed your manuscript and found it of interest. However, there are still concerns need to be addressed before making a final decision for publication.

·

Basic reporting

no comment.

Experimental design

no comment

Validity of the findings

In figure 4, besides Bland-Altman plots, linear regression plot with Pearson’s r2 coefficient should be shown to address the CO and SV measured by the two different methods.

Additional comments

This paper is well designed to address the priority of the DMP-life vs. TTE and showed convincing data for the accuracy of DMP-life, which could be a proper candidate for the hemodynamic detector for the patients. I recommend the author add more statistical graphs like the linear regression plot with Pearson’s r2 coefficient in Fig.4.

·

Basic reporting

Minor comment:
In the exclusion criteria, please confirm the information of LVEF: 40% or 45%?

Experimental design

No comment.

Validity of the findings

The authors of current study well described their work and got the conclusion.

Additional comments

As an alternative choice of pulmonary artery catheter or transthoracic / transesophageal echocardiography measurement, non-invasive radial artery tonometry test has been introduced and functionally well described to measure the aortic pressure, artery stiffness or heart function during the past 20 years. However, less than 20% (107/572) total operated patients could use this method shows that it still need to be well evaluated before widely usage.

Reviewer 3 ·

Basic reporting

no comment

Experimental design

no comment

Validity of the findings

no comment

Additional comments

In this manuscript, Zayat et.al first time compared the DMP-life tonometry device measured SV and CO with TTE in preoperative cardiothoracic surgery patients. Their data indicate reasonable accuracy and precision between two technologies. Manuscript is well written and experiments are straightforwardly designed, but the conclusion needs more data to support. Following is my concerns:

1. Authors only used the Bland-Altman analysis to compare the differences between TTE and DMP-life measurement (both SV and CO). Linear regression analysis to illustrate the correlation efficiency can be considered to compare these two methods.
2. In the discussion section, authors discussed the advantages and disadvantages of TTE and DMP-life. But to better explain the feasibility of DMP-life, authors can compare their data to other reported SV/CO monitoring methods. For example, Castro et al (Castro V De, Lhotel L, Mabrouk N, Perel A, Coriat P. Comparison of stroke volume (SV) and stroke volume respiratory variation (SVV) measured by the axillary artery pulse-contour method and by aortic Doppler echocardiography in patients undergoing aortic surgery. 2006;97: 605–610.) compared SVs measured by the axillary calibrated artery pulse-contour method with TTE, and their percentage error of both minimum and maximum SV was 26%.
3. According to a lot literature, many SV/CO monitoring devices have not yet replaced PACs due to poor trending ability. Authors better to have serial measurements at different hemodynamic states (instead of single measurement) to assess trending ability, and analyze ΔSV values using four-quadrant plot or a polar plot analysis.

Reviewer 4 ·

Basic reporting

The aim of this study is clear that the authors tried to use the non-invasive BPPA to replace PWD method in clinics especially in cardiac surgery patients.

Experimental design

There were some data showed the mean CO and SV were comparable between BPPA and TTE in coronary artery bypass surgery patients. However, there are no data to show the sensitivity and accurate/reliable dynamic range of BPPA.

Validity of the findings

The concept and data in the manuscript are original and useful for clinics. It is much easier to get cardiac and hemodynamic information using this non-invasive way than conventional TTE. Thus, patients will benifit much if this method will be widely used in hospitals.

Additional comments

The authors tried to compare the accuracy and reliability of BPPA used in cardiac surgery patients. Over all, this manuscript is well organized and the data are clearly presented. However, there are some places need to be improved:

1) Figure 1 can be removed from the manuscript. Although it shows some standards that exlude the patients from BPPA test, it does not have any information about the 107 patients enrolled in the test, which should be the key information other than the excluded ones.

2) Figure 2 shows the principle of DMP-life system, however, there were not any explanations in the text. Can the authors add some information to explain the key principle of this system and how it is used in patients.

3) Figure 3 shows the results were got at different applied pressure for each tested patients. What do red rectangles and arrow mean in the figure? The shape are different at different pressure for different persons, so is the shape related to the severity of the heart disease?

5) In Figure 4, the description of the figure legend and the titles in the figure are not consistent. Please check and make the figure in the right order and label.

5) The authors only tested the applicability of DMP-life system in patients with cardiac diseases. Why did not you test it on normal people? so that then we can better know how reliable/accurat it is for this system. I encourage the authors add these data in to make the conclusion more solid.

6) Check the writing and format, and keep them consistent. For example, line 39 and line 193 "5.2±0.85" could be "5.2 ± 0.85".

---

## Round 0.2 · Minor Revisions

Reviewer 3's concern needs to be addressed. Please either explain the correlation results or state it as a limitation of this study.

·

Basic reporting

No comment.

Experimental design

No comment.

Validity of the findings

No comment.

Additional comments

The article is ready for publish.

·

Basic reporting

No comment.

Experimental design

No comment.

Validity of the findings

No comment.

Additional comments

The authors of current study well described their work and got the conclusion.

Reviewer 3 ·

Basic reporting

no comment

Experimental design

no comment

Validity of the findings

no comment

Additional comments

Thanks for the authors performed the pearson correlation efficiency analysis. But the correlation of CO measurements between DMP-life and TTE (r = 0.501) is not a good correlation, which means the CO measurement might be not accurate as TTE. Moreover, authors claimed that this study is to ensure the accuracy of DMP-life device without reproducibility analysis. Then it is very important for authors to include direct comparisons between DMP-life and standard PAC measurement (and can also compared with TTE), especially at this case, the CO measurements didn’t have a significant positive correlation.

Reviewer 4 ·

Basic reporting

The revised manuscript is well improved. The methods, results and discussion are more clearer.

Experimental design

n/a

Validity of the findings

n/a

Additional comments

The authors addressed my concerns and the manuscript was well improved.

---

## Round 0.3 · accepted · Accept

All concerns have been addressed.